# Synergistic Effect of Toceranib and Nanohydroxyapatite as a Drug Delivery Platform—Physicochemical Properties and In Vitro Studies on Mastocytoma Cells

**DOI:** 10.3390/ijms23041944

**Published:** 2022-02-09

**Authors:** Paulina Sobierajska, Anna Serwotka-Suszczak, Sara Targonska, Damian Szymanski, Krzysztof Marycz, Rafal J. Wiglusz

**Affiliations:** 1Institute of Low Temperature and Structure Research, Polish Academy of Sciences, Okolna 2, 50-422 Wroclaw, Poland; s.targonska@intibs.pl (S.T.); d.szymanski@intibs.pl (D.S.); 2Department of Experimental Biology, Faculty of Biology and Animal Science, Wroclaw University of Environmental and Life Sciences, C. K. Norwida 27B, 50-375 Wroclaw, Poland; anna.serwotka-suszczak@upwr.edu.pl (A.S.-S.); krzysztof.marycz@upwr.edu.pl (K.M.)

**Keywords:** nanocarriers, hydroxyapatite, surface properties, targeted release, anticancer drug, mast cell tumour, Toceranib

## Abstract

A new combination of Toceranib (Toc; 5-[(5Z)-(5-Fluoro-2-oxo-1,2-dihydro-3H-indol-3-ylidene)methyl]-2,4-dimethyl-*N*-[2-(pyrrolidin-1-yl)ethyl]-1H-pyrrole-3-carboxamide) with nanohydroxyapatite (nHAp) was proposed as an antineoplastic drug delivery system. Its physicochemical properties were determined as crystallinity, grain size, morphology, zeta potential and hydrodynamic diameter as well as Toceranib release. The crystalline nanorods of nHAp were synthesised by the co-precipitation method, while the amorphous Toceranib was obtained by its conversion from the crystalline form during nHAp–Toc preparation. The surface interaction between both compounds was confirmed using Fourier-transform infrared spectroscopy (FT-IR), ultraviolet-visible spectroscopy (UV–Vis) and scanning electron microscopy with energy-dispersive X-ray spectroscopy (SEM-EDS). The nHAp–Toc showed a slower and prolonged release of Toceranib. The release behaviour was affected by hydrodynamic size, surface interaction and the medium used (pH). The effectiveness of the proposed platform was tested by comparing the cytotoxicity of the drug combined with nHAp against the drug itself. The compounds were tested on NI-1 mastocytoma cells using the Alamar blue colorimetric technique. The obtained results suggest that the proposed platform shows high efficiency (the calculated IC50 is 4.29 nM), while maintaining the specificity of the drug alone. Performed analyses confirmed that nanohydroxyapatite is a prospective drug carrier and, when Toceranib-loaded, may be an idea worth developing with further research into therapeutic application in the treatment of canine mast cell tumour.

## 1. Introduction

Low therapeutic efficacy and multiple side effects are currently two main problems in cancer chemotherapy [1,2]. Nanotechnology shows high potential in cancer treatment as it introduces nanocarriers that can deliver therapeutic agents specifically to tumour sites through enhanced permeability and retention (EPR) effects [3]. Nanomaterials offer multiple benefits in targeted therapies. The physicochemical properties of nanoparticles, such as their size, shape, composition and surface properties, can influence the interaction at the bio-nano interface [4,5]. The large surface area to volume ratio significantly increases the chemical reactions taking place on the surface of nanoparticles. The nanoscale provides an appropriate size for intravascular conveyance and accumulation in tumours [6]. These properties give nanoparticles the potential to target tissues at a molecular level. 

Heretofore, many different systems based on the use of nanoparticles and other nanoscale materials have been proposed in anti-cancer therapies with positive results. Nanohydroxyapatite (nHAp) is one of the widely-studied biomaterials since it demonstrates high bioactivity, biocompatibility, non-immunogenicity, and non-inflammation [7]. As a bioactive material, nHAp interacts with biological entities and thus can form chemical bonds with adjacent biological tissue. In comparison to other inorganic materials, the nHAp is safe to be used since it occurs naturally in bone tissue as well as its biodegradability and biocompatibility are much better than that the other nanoparticles such as SiO_2_, TiO_2_, quantum dots, and carbon nanotubes [8]. Moreover, the nHAp has a low solubility in physiological pH and an even higher stability than liposomes or micelles [9]. In its mesoporous form, the hydroxyapatite is able to adsorb larger amounts of therapeutic molecules and provide a prolonged release of active substances [10,11]. Many studies have described the versatility of this material in combination with other compounds and as drugs conjugates with implanted scaffolds or polymer-coated nanoparticles [12,13,14]. Some recent studies identified nanohydroxyapatite as the preferred material for drug carrier design [7,15,16,17,18,19,20]. Synthetic nHAp are of great interest in biomedicine as an orthopaedic and dental implant material in the form of nanoparticles, nanocoatings, porous scaffolds, or nanocomposites [21]. Nanocrystalline hydroxyapatite shows no post-implantation cytotoxic effect, enhances osteoinductivity and osteointegration [22]. Compared to other inorganic nanocarriers, the nHAp achieves many delivery efforts in delivering sustained-release anti-tumour agents and is biodegradable in body fluids [23,24]. Zuo et al. investigated the hydroxyapatite/methotrexate (HAp/MTX) hybrids with different morphologies toward tumour cell inhibition against HeLa cell line [25]. Fan et al. demonstrated, in a tumour-bearing mouse model, excellent tumour growth inhibition using doxorubicin-loaded hydroxyapatite nanorods consisting of folic acid modification (DOX@HAP-FA) compared with the free DOX [26]. Martins et al. shown that after 72 h, even a low level of chemotherapeutic paclitaxel (PTX) in combination with nHAp can reduce the cells’ viability (Breast cancer cells, MCF-7 lineage) by stimulating the apoptotic phenotype and suppressing of survival stimuli [27]. The studies by Talegaonkara et al. clearly indicated the synergistic anti-cancer activity of the Bendamustine Hydrochloride-loaded nHAp compared to the drug itself, caused by nanoparticle internalization by the cells and the intracellular Ca^2+^ release inducing tumour proapoptotic mechanisms (in-vitro, ex-vivo and in-vivo tests) [28]. Chen et al. after analysis of mRNA and protein found that recombinant mutant human tumour necrosis factor α (rmhTNF-α) inhibited the gene expression of XIAP, survivin, Ki67, PCNA, MDR1 and BCRP as good or better when the nHAp was incorporated in chemotherapy of multidrug-resistant hepatocellular carcinoma [29]. 

It seems that the modification of active compounds with nanohydroxyapatite as a nanoscale drug carrier may be a promising prospect. Our previous results regarding nHAp modified with Imatinib showed that this combination enhanced the effectiveness of the attached drug (led to a significant reduction in the viability of NI-1 cell line) [30]. In line with the idea of searching for a new anticancer drug conjugation with nanocrystalline hydroxyapatite, this article describes the effect of Toceranib in combination with the nanohydroxyapatite (nHAp-Toc) as an effective therapeutic tool against mastocytoma.

Since the selection of nanoparticle material is mainly dictated by the type of drug to be delivered, we decided to use Toceranib (5-[(5Z)-(5-Fluoro-2-oxo-1,2-dihydro-3H-indol-3-ylidene)methyl]-2,4-dimethyl-*N*-[2-(pyrrolidin-1-yl)ethyl]-1H-pyrrole-3-carboxamide). Its salt, Toceranib phosphate is commercially used as Palladia^®^ (Pfizer Animal Health, Madison, NJ, USA) in the treatment of canine mast cell tumour (MCT). Therefore it seems to be an interesting solution to combine Toceranib with a phosphate platforms such as calcium phosphates (CaPs) [31]. Synthetic nanohydroxyapatite (Ca_10_(PO_4_)_6_(OH)_2_), which belongs to the group of CaPs, exhibits excellent biocompatibility and could be successfully used as a drug carrier for Toceranib to improve the bioavailability of chemotherapeutic [32,33]. Toceranib is a specific small-molecule tyrosine kinase inhibitor with both direct anti-tumour and anti-angiogenic activity, mechanism of which lies in competitive blocking of adenosine triphosphate binding domain [34]. Toceranib selectively inhibits the activity of receptor tyrosine kinases (RTK), many of which are associated with tumour growth, pathological angiogenesis, and tumour progression through metastasis. It inhibits the activity of the tyrosine kinase Flk-1/KDR (vascular endothelial growth factor receptor, VEGFR2), platelet-derived growth factor receptor (PDGFR) and the stem cell growth factor receptor (c-Kit), which has been observed both in biochemical and cellular studies [35]. In vitro studies have shown that it exerts an anti-proliferative effect on endothelial cells. Toceranib induces cell cycle arrest and consequent apoptosis in tumour cells expressing activation of mutations in the RTK kinase, c-Kit. In 2009, London et al. concluded that Toceranib serves as an effective single agent against MCTs in dogs [23]. Canine mast cell tumours, as the most common skin cancer in dogs (representing about 20% of cases), are often caused by activating a mutation in c-Kit [36]. Toceranib exhibits significant biological activity against MCT in dogs, the response rate to single-agent being about 42.8% seems to be favourable compared to that observed with other single-agents and their combinations [35]. Moreover, clinical trials have shown that spontaneous tumours in dogs are good models for assessing the therapeutic index of targeted therapeutic agents.

For all of the above-mentioned reasons, we believe this drug may be a good model for testing nHAp as a drug delivery platform. To the best of our knowledge, it is the first work that comprehensively describes the physicochemical properties of nHAp-Toc composite and related cell-culture studies in vitro.

## 2. Results

### 2.1. Characterisation of nHAp, Toc and nHAp-Toc

The XRPD (X-ray powder diffraction) was used to characterise phase crystallinity and purity of the hydroxyapatite (nHAp), Toceranib (Toc) and the nHAp-Toc composite. The results have been gathered in Figure 1. The XRPD pattern of the obtained nanohydroxyapatite (green line) presents the pure hexagonal phase where the peaks correlate with those derived from the theoretical pattern of the hydroxyapatite no. ICSD (Inorganic Crystal Structure Database)-26204 (yellow line) [37]. In the case of nHAp-Toc (red line), reflection planes corresponding to the hydroxyapatite host lattice were also detected, which confirmed that nHAp was not altered by the combination with the drug. Several distinctive peaks of the nHAp were detected at 2θ equal to 26.0°, 31.9°, 32.3°, 33.0°, 34.2°, 39.7°, 46.7°, 49.5° and 53.1°. The unit cell parameters and crystallite sizes (~46 nm) for this material were calculated and presented in our previous work, thus the nano-nature of the obtained nHAp was confirmed [30]. 

The two different polymorphs of Toceranib (5-[(5Z)-(5-Fluoro-2-oxo-1,2-dihydro-3H-indol-3-ylidene)methyl]-2,4-dimethyl-*N*-[2-(pyrrolidin-1-yl)ethyl]-1H-pyrrole-3-carboxamide) are summarised in Figure 1, crystalline (black line) and amorphous (blue line) forms. For crystalline form (the material directly obtained from the manufacturer), there was found a series of sharp lines at 2θ equal to 12.3°, 13.8°, 14.3°, 16.0°, 17.4°, 19.0°, 20.9° and 24.8°. However, the amorphous phase of Toc (obtained under the same conditions as for the preparation of the composite) is observed as a broad band in the range of 2θ between 11°–29°. This amorphous halo is clearly also visible for the nHAp-Toc diffraction pattern (red line), which proves the interaction of crystalline hydroxyapatite with an amorphous drug. The same effect, i.e., the transformation of the crystalline form of the drug into an amorphous one during the preparation of the nHAp-Toc, was found in our previous work on another antineoplastic drug—Imatinib [30].

The use of the FT-IR (Fourier-transform infrared spectroscopy) technique may be helpful in considering how Toceranib may interact with nanohydroxyapatite. The FT-IR spectra of nHAp, Toc, as well as their composite have been gathered in Figure 2. The spectrum of the nHAp (green line) shows the typical peaks of PO_4_^3−^ group located at 1044 cm^−1^ and 1095 cm^−1^ and attributed to the triply degenerated antisymmetric bending vibration ν_3_ of the P–O bond [38]. The peaks at 564 cm^−1^ and at 604 cm^−1^ are characteristic for the antisymmetric bending modes ν_4_ of the phosphate groups. Bands at 961 cm^−1^ arise from symmetric stretching modes ν_1_ of PO_4_^3−^ groups. The vibration frequencies observed at 3573 cm^−1^ and 633 cm^−1^ were attributed to the stretching and bending modes of OH^−^ groups, respectively. The presence of structural –OH groups built into apatite crystal is clearly confirmed by the appearance of those two characteristic frequencies [39]. 

The FT-IR spectra of Toceranib (black line) as well as nHAp-Toc (red line) display numerous characteristic frequencies marked in Figure 2. The peaks at 2964 cm^−1^ and at 2868 cm^−1^ were attributed to the –C-H stretching vibration of methyl groups, antisymmetric and symmetric, respectively. At 1477 cm^−1^, lines corresponding to the antisymmetric bending vibration of –C-H bond were detected. The spectral range between 1200 cm^−2^ and 800 cm^−1^ was built by absorption peaks associated with stretching vibrations of the C-C bond. The bending vibration of the C-C bond appeared at 448 cm^−1^. Peaks in the range of 900 cm^−1^–650 cm^−1^ were correlated to out-of-plane C-H bending vibrations [40]. The characteristic frequencies attributed to C=O vibration were observed at 1675 cm^−1^ [41], while the N-H stretching bands were found at 3194 cm^−1^ [42]. The absorption band at 3390 cm^−1^ was related to the hydrogen-bonded -NH groups [12].

All these absorption peaks for nHAp-Toc composite indicated the existence of surface interaction between both components, the nanohydroxyapatite and Toceranib, showing that after loading the nanoparticles with drug, nHAp retained its crystal structure.

Zeta potential (ζ) characterise the surface of charged suspension of particles and can thus affect the pharmacokinetic properties of drug-nanocarrier systems in body fluids [43]. Small changes in zeta potential were detected after modification of nHAp in the cell culture medium (Figure 3, inset), ranging from −10.4 to −11.7 (mean values). These results show that Toceranib had no significant effect on the surface charge of nHAp. Additionally, the typical hydrodynamic sizes of nHAp and nHAp-Toc were estimated to be around 255 nm and 396 nm, respectively. They are quite large and indicate that the material tends to agglomerate in the tested medium due to the nano-size (high surface area) of the nHAp particles. The tendency towards agglomeration was also checked by the SEM (scanning electron microscopy) technique. 

The morphology of nHAp-Toc sample was investigated by using SEM. As shown in Figure 4, the particles of nHAp exhibit an exceptional tendency to aggregate into irregular cluster shapes (up to several microns in size) which are covered by the amorphous phase of Toceranib. According to our previous work [30] we proved with the SEM method that the observed aggregate is composed of the nanometric HAp particles with elongated rod-like shapes. The phenomenon of interaction between nHAp and Toc was also confirmed by elemental maps of carbon (C), calcium (Ca) and phosphorus (P). The element distribution of C (carbon atoms were used as the marker of the drug) clearly shows that Toceranib efficiently covers the agglomerated nHAp particles (see SEM image and elements distribution of Ca or P, Figure 4). From the EDS (energy-dispersive X-ray spectroscopy) maps it is also visible that the all elements are uniformly distributed throughout the nHAp-Toc sample. A similar observation about the interaction between nHAp and drugs was reported in our previous work [30].

The results of the XRPD, FT-IR, and SEM-EDS measurements show that the nanohydroxyapatite and Toceranib interact by surface adsorption but do not indicate any significant chemical bonds between these two components.

The UV-Vis (ultraviolet–visible spectroscopy) absorption spectra of Toceranib have been shown in Figure 5A (inset). The concentration of drug in the nHAp-Toc was estimated using the absorption calibration curve (Figure 5A). The calculated concentration of Toceranib (Analyte) was 86 μg/mL. The estimated value of drug-loading capability (LC) was 2 wt%.

Figure 5B presents the release profile of Toceranib from the nHAp-Toc in PBS (phosphate-buffered saline) medium at pH 7.4. The burst release in the early stage is noticeable. About 16.7% of Toc was released in the first 5 min. Over the next 6 h, only an additional 2.2% of the drug was released. Then, for the following hours the amount of drug in the medium increased slightly and reached a total value of 22.7% after 24 h. This result is interesting because it is significantly different from that of the nHAp-Imatinib in our previous work [30] (Imatinib was completely released after 45 min incubation in PBS). First, the possible stronger interactions of the nHAp surface with the Toc should be considered (especially since Toceranib is also present as a phosphate salt). Second, unlike Imatinib, Toceranib is less soluble in aqueous suspensions, resulting in less drug being dissolved. Third, the released Toceranib reacts with the phosphate ions contained in PBS and precipitate as its phosphate salt. It should be emphasised that by reducing the pH of the environment towards acidic pH, the drug is effectively released, and the hydroxyapatite is dissolved at acidic pH. It should also be noted that nHAp itself undergoes constant remodelling and resorption in the body, which also affects the drug release profile, thus increasing its bioavailability. On the other hand, the slow drug release may be beneficial due to the possibility of eliminating the side effects of the drug, related to the sudden increase in the amount of the active substance in body fluids. 

### 2.2. Cytotoxicity Evaluation of the nHAp-Toc

To check the effect of the combination of Toceranib with nanohydroxyapatite on drug efficacy, the cytotoxicity of the tested compounds was investigated using the Alamar blue colorimetric technique. A range of 8 different concentrations of Toceranib alone and with the nanohydroxyapatite (1 × 10^−11^ to 2.29 × 10^−5^ M for Toc and 1 × 10^−10^ to 1 × 10^−3^ g/mL for nHAp) was used to assess the effect of experimental compounds, using the TOX-8 assay after 48 h of incubation.

Figure 6 shows the results of the cytotoxicity analysis of the tested compounds. The charts were prepared using GraphPad Prism 5.01 software (GraphPad Software Inc., San Diego, CA, USA). The half maximal inhibitory concentration coefficient IC_50_ (as well as the R^2^ determination coefficient) was noted where such calculations could be made. This factor is a measure of the potency of a substance to inhibit a specific functions (biological or biochemical), in this case it stands for the metabolic activity of cells.

Two control cell lines for the experiment were used in the study. The canine osteosarcoma cell line D17 was selected as the control line due to the absence of a mutation in the gene c-Kit. The L929 mouse fibroblast cell line was taken as another control line in the experiment, since it is a normal (non-cancer) cell line without any known mutation in receptor tyrosine kinases. The results obtained for control line D17 are as expected, no effect of any of the test substances on the viability of the cells was observed (Figure 6G–I). However, for the second control cell line tested, the L929 mouse fibroblasts, a decrease in viability can be seen as the concentrations of the Toceranib increase, both alone and in combination with nHAp (Figure 6D,E). Nevertheless, it should be noted that the cell viability observed does not fall below 50%. The observed cytotoxicity may be the result of significantly high concentrations of the tested drug used in the experiment. An important result is also the lack of influence of nHAp alone on the metabolic activity of cells of both control cell lines (Figure 6F,I), which is consistent with the previously described biocompatibility of nanohydroxyapatite [23]. Figure 7B,C represent the comparison of the effect of Toc alone and in combination with the nHAp for both control cell lines. It confirms that that the difference in observed influence is minimal.

The model line selected for the experiment was a canine mastocytoma NI-1 cell line, used for drug resistance testing with confirmed c-Kit tyrosine kinase mutations [44]. The results obtained with this cell line turned out to be very promising. Toceranib exhibits concentration dependent cytotoxicity, as expected (Figure 6A). In our experiment, we obtained an IC_50_ of 5.28 nM, which is very similar to that described for other cell lines (6–10 nM) [45,46,47]. The result obtained for the nHAp-Toc (Figure 6B) is interesting, and the calculated IC_50_ is even lower (4.29 nM) than that obtained with the drug alone. It is worth noting that the decrease in viability in this test is much more noticeable at higher concentrations than for the drug alone (Figure 7A). Cells of this line appear to be more sensitive to nanohydroxyapatite alone (Figure 6C); at very high concentrations a decrease in viability is seen more than in the control lines, but cell viability still does not drop below 50% even at significant concentrations.

## 3. Discussion

In this study, the nanohydroxyapatite particles loaded with Toceranib were developed for use in cancer targeted therapies. Their combination was aimed at achieving a synergistic effect of both components of the composite which results in increasing the effectiveness of such a system while reducing the active dose of the drug. This approach presents a particular challenge as currently used cancer therapies are highly unpredictable, treatment efficacy is difficult to achieve and patients suffer from severe side effects. In our experiment, it has been found that both different forms of components have a positive effect on the bioavailability of the drug: the crystalline phase of the nHAp increases surface area for dissolution and the amorphous form of Toceranib dissolves faster than its crystalline forms due to its high free energy and low density [48]. In this concept, multi-state component systems deserve special attention. Our results are in line with medical market expectations—the need to enhance the solubility of active drug substances without affecting their stability or pharmacological activity [49]. Here, in reference to the XRPD and SEM-EDS results, the co-existence of both phases was found. Therefore, in this case the synergistic therapy includes a combination of nanocrystalline inorganic material with an amorphous drug. 

The development of nanocarriers in drug delivery system seems to be highly attractive for chemotherapy. According to the needs of clinical researches, the drug-loaded nanocarriers show high targeting, controlled release capability and improved cell penetration ability [50]. Moreover, they can extend half-life of drug and improve its efficacy while reducing side effects of medication. The biggest feature that distinguishes nanoparticles from other materials is their high surface to volume ratio, bringing the efficiency of attaching drug to the nanocarrier resulting into better treatment [8]. The large surface area of the nanoparticles facilitates the incorporation of drugs and, when applicated, increases the reactivity with cell membranes (including penetration, binding and modification of proteins), which enhances the effect of drug delivery to a specific site [51]. However, several questions about the safety of using such small objects remain open. Undoubtedly, their toxicity depends on the route of administration and exposure [52]. An example may be given metal oxide nanoparticles that can cause oxidative stress, inflammation, and DNA damage [53,54,55,56]. One of the effective strategies to avoid potential risks can be the selection of biodegradable nanomaterials with high biocompatibility. Among the available nanomaterials, the crystalline nanohydroxyapatite (nHAp) meets these criteria and has been approved by the FDA (Food and Drug Administration, Silver Spring, MD, USA) for application as a bone grafts [57]. Our experiment confirms the biocompatibility of the obtained nHAp, which is essential for its medical use. This is in accordance to previous findings on cytotoxicity evaluation of nHAp-based materials [58,59,60,61]. 

Cancer treatment is a formidable challenge. Several nanocarriers have been applied in cancer chemotherapy, such as liposomes (Doxil^®^, Lipusu^®^), nanoparticles (Abraxane^®^, Feraheme^®^) and micelles (Genexol-PM^®^, CriPec^®^) [57,62]. A large number of inorganic nanoparticles (NPs) have been extensively studied for chemotherapy (e.g., graphene oxide-based NPs (GO NPs) [63,64], mesoporous silica NPs (MSN NPs) [65], gold NPs (AuNPs) [66], and iron oxide NPs (IO NPs) [67]). Many studies showed synergistic effect of drugs and nanoparticles in vitro and in vivo [68,69,70]. Although, some of them describe the use of nanohydroxyapatite (nHAp) as a biocompatible anti-cancer delivery vehicles (see examples in the Section 1), data on this issue are rather insufficient. In this report, the new nHAp-mediated delivery system for Toceranib (Toc) has been proposed. In our research, we decided to use a Toceranib (protein tyrosine kinase inhibitor), drug with a known mechanism of action, marked effectiveness, already approved for use in animals in mast cell cancer. 

As described in the data from the available literature, the main drug-loading mechanisms for inorganic porous materials as carriers are noncovalent electrostatic, π–π stacking, hydrogen bond, and hydrophobic interactions [8]. Combining the data from FT-IR, UV-Vis and SEM-EDS, the surface interaction of nHAp with Toc has been confirmed. Moreover, it has been shown that ~23% of the drug is released into the biological medium used (pH = 7.4). On the contrary, Imatinib release from the nHAp achieves 100% within the same time period as we have shown in our previous work [30]. Referring to the current article, similar release profiles was observed for DOX@HAP-FA system, where the doxorubicin (DOX) release from the nHAp nanorods at pH 7.4 was only ~16% (24 h of incubation in PBS) [26]. The authors concluded that this provides the ability to enhance the availability of the drug to tumours. Subsequent reabsorption and remodelling of the nHAp in the body weaken the interaction between nanoparticles and Toceranib, which favours the effective release of the drug related to the pH drops. This phenomenon has been described in several papers [26,71,72]. Although, the value of estimated drug-loading capability is low, the strategy to functionalization nanocarrier first followed by drug loading could achieve high drug-loading nanoparticles [73]. An example can be given the functionalization of nHAp surface by alendronate (ALN), using ibuprofen (IBU) as a model drug [74]. In this concept, due to the ionic interaction between –NH_3_^+^ and –COO^−^, a high IBU storage capacity was obtained and relatively favourable drug release profile was achieved compared to pure nHAp.

The size, shape, charge and surface chemistry of nanoparticles are crucial parameters in relation to the cellular uptake mechanisms [75]. Despite the tendency to agglomerate nanoparticles with the drug demonstrated by large hydrodynamic radius and zeta potential of about −12 (particles with potential of about ±30 are normally considered stable [76]), cytotoxicity tests have shown that the nHAp-Toc system has the potential (in vitro studies) to treat canine mast cell (MCT) tumours. The results obtained in in vitro tests are promising. In our research, we have shown that the combination of this drug with nanohydroxyapatite may, to a small extent, positively affect its effectiveness, while maintaining the specificity of the drug alone, which is proven by the lack of impact on the control lines. These results are encouraging, as they may give rise to research that would allow us to assess whether it is possible to use the drug in lower doses in the nHAp-Toc system or whether this combination will reduce the drug resistance effect described for Toceranib in the literature as a result of prolonged exposure [46,77].

## 4. Materials and Methods

### 4.1. Synthesis of nHAp 

The nanohydroxyapatite (nHAp) was prepared according to the following procedure. Stoichiometric amounts (10 mmol) of Ca(NO_3_)_2_·4H_2_O (99+% Acros Organics, Geel, Belgium) and (6 mmol) of (NH_4_)_2_HPO_4_ (≥98% Avantor Performance Materials Poland S.A., Gliwice, Poland) were taken and subsequently dissolved under constant stirring in separate glass beakers containing MQ-water. Afterwards, the (NH_4_)_2_HPO_4_ solution was added dropwise into the calcium nitrate one, and fast precipitation of the pre-product was directly observed. After all the phosphate-containing solution has been added, NH_3_∙H_2_O was used as a pH-stabilizing agent in order to keep the alkaline reaction (pH = 10). The mixture was aged for 3 h. Subsequently, white precipitate was separated using fast laboratory centrifuge, washed with MQ-water until neutral pH, and then dried to a powder. The last step of the synthetic procedure involved thermal treatment of the nHAp particles at 500 °C for 3 h with the aim relying on removal of NH_4_NO_3_ and complete crystallisation of amorphous content. The resulting white powder were directly used for preparation of composite with the anti-cancer drug.

### 4.2. Modification of nHAp by Toceranib (nHAp-Toc)

The surface of nanohydroxyapatite was modified by Toceranib ((5-[(5Z)-(5-Fluoro-2-oxo-1,2-dihydro-3H-indol-3-ylidene)methyl]-2,4-dimethyl-*N*-[2-(pyrrolidin-1-yl)ethyl]-1H-pyrrole-3-carboxamide), ≥98% (HPLC) Sigma Aldrich, St. Louis, MO, USA). The chemotherapeutic was dissolved in DMSO (≥99.9% Sigma Aldrich, St. Louis, MO, USA) until complete dissolution, and was then added to the nanohydroxyapatite water colloidal suspension and was sonicated for 2 h. The dispersion was incubated and further diluted in a complete cell culture medium.

The concentration of nHAp in the stock solutions was 500 µg/mL for the nHAp-Toc, while the Toceranib concentration in the stock solution equal to 100 µg/mL.

### 4.3. Physicochemical Characterization

The XRPD patterns were collected in the range from 5 ° to 80 ° with PANalytical X’Pert PRO X-ray diffractometer, (Malvern Panalytical Ltd., Royston, UK) equipped with Ni-filtered Cu Kα1 radiation (Kα1 = 1.54060 Å). The measurements were done under the conditions: voltage: 40 kV, current: 30 mA, scan angle (2θ) between 5° and 80° (step size = 0.0263°, time per step = 2.5 s). The FT-IR spectra were measured with a Thermo Scientific Nicolet iS50 FT-IR spectrometer (Thermo Fisher Scientific, Waltham, MA, USA) equipped with an Automated Beamsplitter exchange system (iS50 ABX containing DLaTGS KBr detector), in the 4000–400 cm^−1^ region using KBr pellets at room temperature, and the background noise was corrected with pure KBr data. The HeNe laser was used to generate an infrared radiation source. The SEM images and EDS maps of the nHAp-Toc were acquired using the Field Emission Scanning Electron Microscope (FEI Nova NanoSEM 230, FEI Company as a part of Thermo Fisher Scientific Inc., Hillsboro, OR, USA) equipped with an EDS spectrometer (EDAX Genesis XM4). The sample of the material was dispersed in ethanol and a drop was placed on the silicon stub. After evaporating the solvent (using infrared radiation), the specimen was put under the microscope. In the SEM measurements, the sample was analysed by electron beam at a low accelerating voltage of 5.0 kV and using a secondary electron (SE) detector. In the case of SEM-EDS measurements, the sample was scanned at 10.0 kV using EDAX Apollo X Silicon Drift Detector (SDD) along with EDAX Genesis Software. The absorption spectra were recorded by using Agilent Cary 5000 UV–Vis–NIR spectrophotometer (Agilent Technologies, Santa Clara, CA, USA) employing spectral bandwidth of 0.1 nm in the ultraviolet-visible (UV-Vis) range. The spectra were recorded from 230 nm to 450 nm (43,478–22,222 cm^−1^). The concentration of Toceranib in the nHAp-Toc was calculated from the calibration curve of the drug. The hydrodynamic size and Zeta potential of the obtained materials in cell-culture medium were analysed by DLS and Zeta potential analyser Zetasizer Nano ZS apparatus from Malvern Instruments (Malvern, UK), operating under He-Ne 633 nm laser and equipped with the Dispersion Technology Software. Each measurement was repeated three times. The hydrodynamic radius (rh) and zeta potential (ζ) of the particles in prepared suspensions were estimated with the Stokes–Einstein [10] and Henry’s [78] equations, respectively.

### 4.4. Toceranib Content and Release of the Drug from nHAp-Toc

The Toceranib content in the nHAp-Toc was calculated form the UV-Vis (λ_m_ = 435 nm) calibration curve based on a series of drug dilution (0 up to 12.5 μg/mL) at room temperature in DMSO solution. The drug-loading capability (LC) was evaluated by determining the total amount of Toc, free drug in the supernatant, and nanohydroxyapatite weight according to the equation: (1)$$LC=\frac{W_{drug}}{W_{drug}+W_{nanparticles}}\times 100\%$$

The drug release was conducted by the ultracentrifugation (12,500 rpm, 5 min) technique in PBS buffer (pH 7.4) at 37 °C. The supernatant was sampled at fixed intervals: 0, 5, 10, 30 min as well as 1.5, 6, 18 and 24 h, and was analyzed determining the Toceranib concentration with UV-Vis detection (λ_m_ = 435 nm). The standard solutions were prepared with PBS buffer.

### 4.5. Cell Cultures

Canine mastocytoma NI-1 cell line was kindly donated to us by Dr. med. vet. Emir Hadzijusufovic from the Clinic for Internal Medicine and Infectious Diseases, University of Veterinary Medicine Vienna, Vienna, Austria. NI-1 cells were grown in suspension in RPMI-1640 medium (Sigma-Aldrich, St. Louis, MO, USA), supplemented with 10% foetal bovine serum (FBS). Control cell lines L929 murine fibroblasts and D17 canine osteosarcoma cells were selected. L929 cells (American Type Culture Collection, ATCC, Manassas, VA, USA) were cultured in Minimum Essential Medium Eagle (MEM, Sigma-Aldrich, St. Louis, MO, USA) containing 10% FBS (Sigma-Aldrich, St. Louis, MO, USA), while D17 cells (European Collection of Authenticated Cell Cultures, ECACC, Salisbury, UK) were cultured in modified Eagle’s medium (MEM) supplemented with F12 Ham’s nutrient and 10% FBS. Cells were cultured in standard conditions (humidified incubator, 37 °C, 5% CO_2_).

### 4.6. Cytotoxicity Assay

Alamar blue colorimetric assay (TOX8 In Vitro Toxicology Assay Kit, Sigma-Aldrich, St. Louis, MO, USA) was used to analyse the cytotoxic effect of Toceranib alone, unmodified nHAp and the nHAp-Toc combination. In this technique metabolically active cells reduce resazurin, which is then measured colorimetrically. The procedure was performed in the same way for all cell lines, according to the protocol provided by the manufacturer. Briefly, 5000 cells per well in 100 µL of dedicated medium were seeded on 96-well plates and cultured overnight in standard conditions. Then, cells were incubated with Toceranib, modified and unmodified nHAp at concentrations from 1 × 10^−11^ to 2.29 × 10^−5^ M (Toceranib) and 1 × 10^−10^ to 1 × 10^−3^ g/mL (nHAp) for 48 h under standard conditions. Afterwards, the media from cell cultures were replaced with 10% *v*/*v* TOX8 dye solution in full medium for 2 h and the spectrophotometric measurement was carried out using a microplate reader (Epoch BioTek^®^, Winooski, VT, USA) at 600/690 nm wavelengths. A full medium supplemented with 10% *v*/*v* dye solution was used as blank. Statistical analysis was determined using GraphPad Prism 5.01 (San Diego, CA, USA). Statistical significance was analysed by unpaired *t*-test and *p* values < 0.05 were considered as statistically significant. The results shown in the figures represent mean values ± standard deviation (SD). P values less than 0.05 (*p* < 0.05), *p* < 0.01 and *p* < 0.001 were summarised with one (*), two (**) or three asterisks (***), respectively.

## 5. Conclusions

Looking beyond nHAp-drug interactions, either chemically or physically, the ability of HAp nanoparticles to react with other functional groups, with plasma membrane components (phospholipids, oligosaccharides, proteins etc.), or extracellular matrix proteins that alter or modulate cellular responses (activate or inhibit cell signalling pathways, e.g., proliferation, differentiation, adhesion, production of reactive oxygen species, communication, apoptosis, etc.) are still an insufficiently understood. Nevertheless, basic physicochemical research provides an important contribution to understanding these processes. To the best of our knowledge, the nHAp-Toceranib composite has been fabricated and comprehensively described for the first time. The results obtained from biological experiments are very promising and constitute preliminary studies allowing researchers to assess the possibility of minimizing the dose of the drug used in conjunction with nHAp, as well as to assess reduced drug resistance.

## Figures and Tables

**Figure 1 ijms-23-01944-f001:**
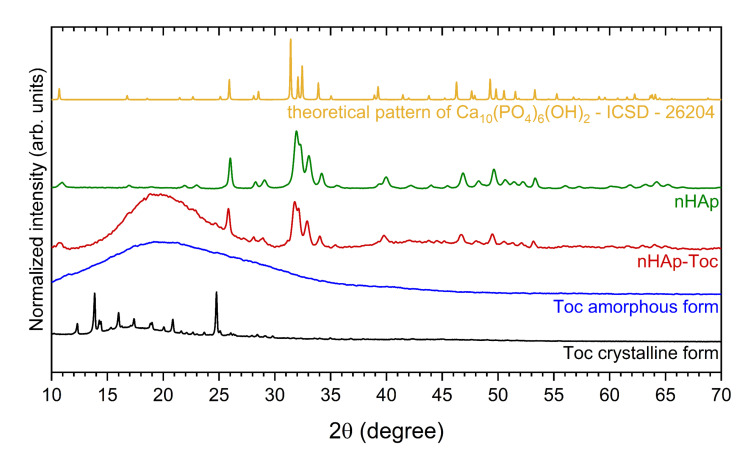
X-ray powder diffraction (XRPD) patterns of the crystalline Toceranib (Toc, black line) and amorphous Toceranib (Toc, blue line), the nanohydroxyapatite modified with Toceranib (nHAp-Toc, red line) and the nanohydroxyapatite (nHAp, green line) together with the reference hydroxyapatite pattern (ICSD—26204, yellow line).

**Figure 2 ijms-23-01944-f002:**
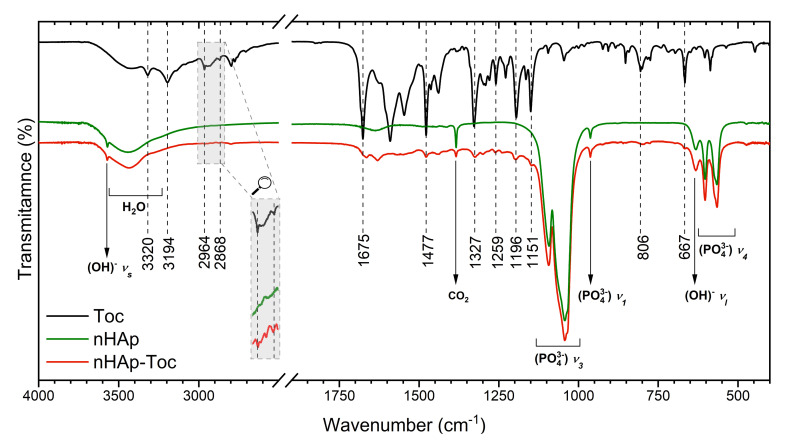
Fourier-transform infrared (FT-IR) spectra of the Toceranib (Toc, black line) nanohydroxyapatite (nHAp, green line) and nanohydroxyapatite modified with Toceranib (nHAp-Toc, red line).

**Figure 3 ijms-23-01944-f003:**
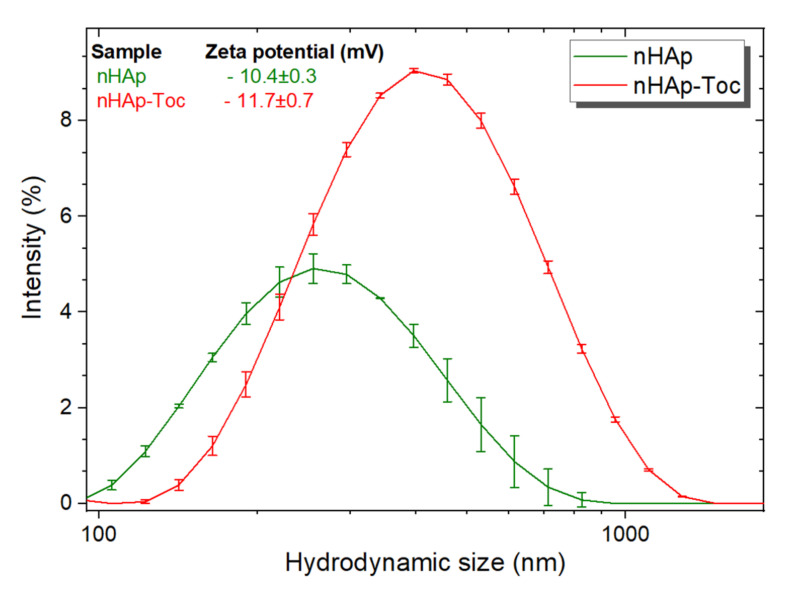
Hydrodynamic size of the nanohydroxyapatite (nHAp) and nHAp modified with Toceranib (nHAp-Toc) monitored in cell culture medium, together with the values of zeta potentials (inset).

**Figure 4 ijms-23-01944-f004:**
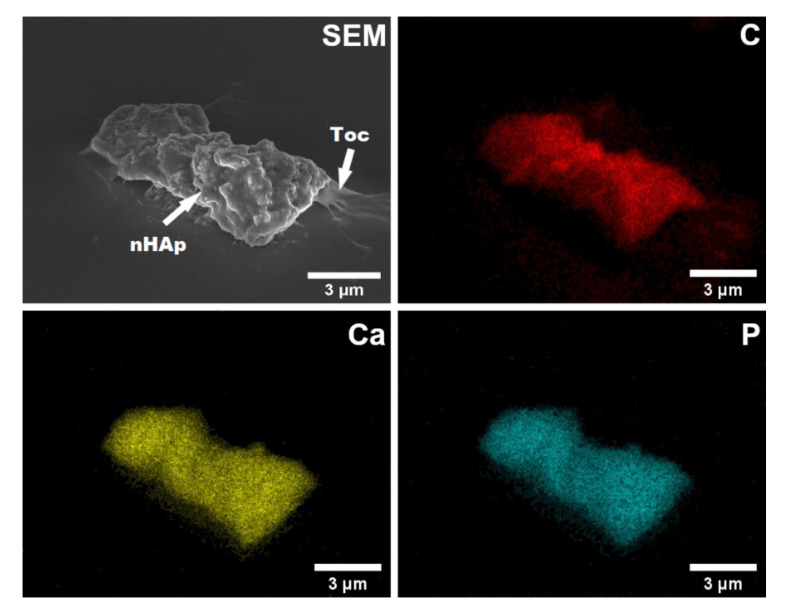
Representative scanning electron microscope (SEM) image and elemental mapping of carbon (C), calcium (Ca) and phosphorus (P) by energy-dispersive X-ray spectroscopy (EDS) in the nHAp-Toc.

**Figure 5 ijms-23-01944-f005:**
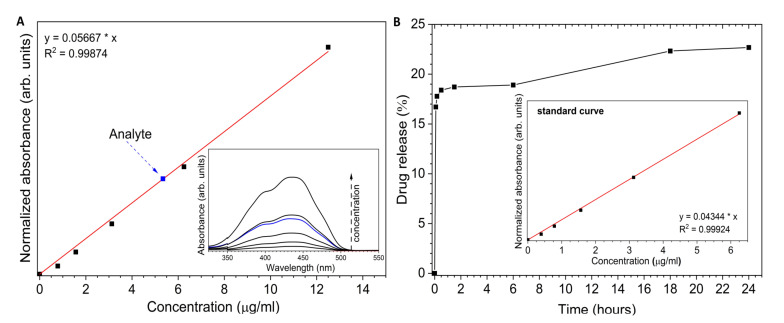
(**A**) Determination of the concentration of Toceranib (analyte) in nHAp-Toc (16-fold dilution of the stock solution). (**B**) Time dependent of the Toceranib release from the nHAp-Toc.

**Figure 6 ijms-23-01944-f006:**
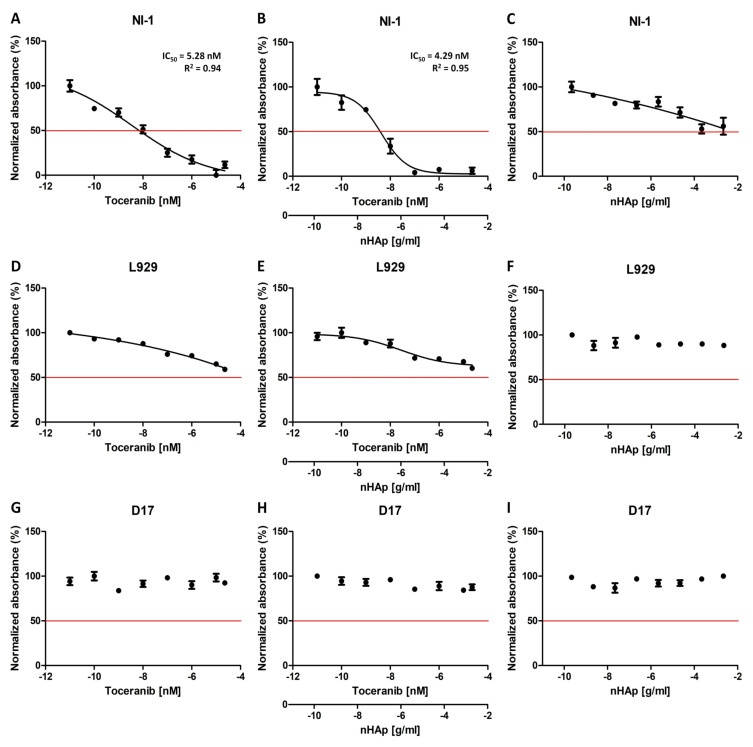
Cytotoxicity test (Alamar blue) results of the Toceranib, nHAp-Toc and nHAp alone on NI-1 (**A**–**C**), L929 (**D**–**F**) and D17 (**G**–**I**) cells. IC_50_ is indicated where possible. R^2^ is the coefficient of determination.

**Figure 7 ijms-23-01944-f007:**
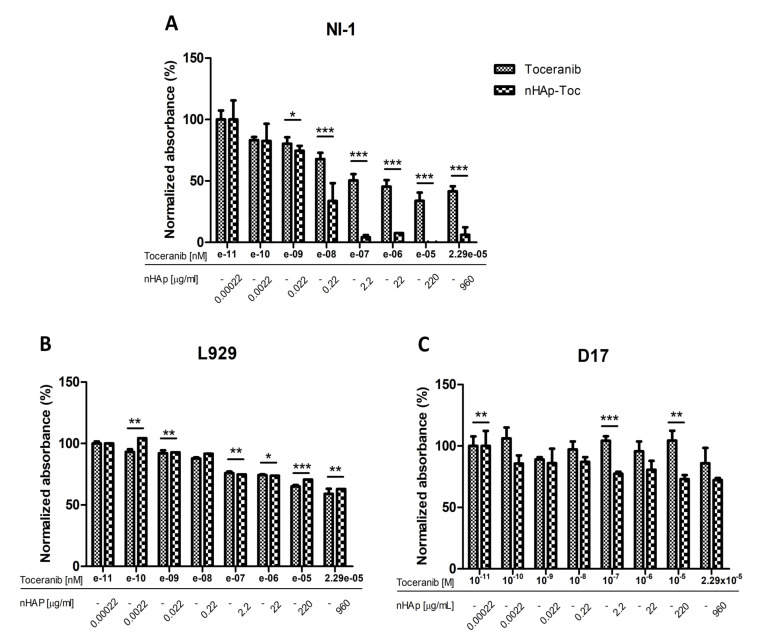
Juxtaposition of the results of cytotoxicity analysis of Toceranib and nHAp-Toc on NI-1 (**A**), L929 (**B**) and D17 (**C**) cells. Results are expressed as mean ± SD, * *p* < 0.05, ** *p* < 0.01, *** *p* < 0.001.

## Data Availability

Data are available from the authors upon request.

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
