# Peer review of "Synergistic Effect of Toceranib and Nanohydroxyapatite as a Drug Delivery Platform—Physicochemical Properties and In Vitro Studies on Mastocytoma Cells"

_ijms, 2022, doi:10.3390/ijms23041944_

Round 1
Reviewer 1 Report
The manuscript is based on the authors’ preliminary studies regarding nHAp modified with Imatinib, in which this combination enhanced the effectiveness of the attached drug against NI-1 cell line. The introduction is written in a cohesive manner and the in-vitro biological evaluation is appreciated. However, I find it difficult to understand this present work without first reading the previous one, which might represent a problem for the reader. For instance, many of the comments below were addressed in doi:10.3390/molecules25204602 and further assumed that all drugs perform in an analogous manner. Please find below my specific comments:
- Section 1 (L80-84): Please better express this idea. It might confuse the reader, since later on in this experiment the authors used Toceranib ((5-[(5Z)-(5-Fluoro-2-oxo-1,2-dihydro-3H-indol-3-ylidene)methyl]-2,4-dimethyl-N-[2-(pyrrolidin-1-yl)ethyl]-1H-pyrrole-3-carboxamide), ≥ 98% (HPLC) Sigma Aldrich, USA) and not its phosphate salt.
- In L 215 there is another reference to this fact…one should understand that the Toc reacts with HAp and transforms to its phosphate salt?
- Section 2.1: Where did the authors get/isolate the amorphous phase to analyze by XRD? From the abstract one can understand that the “amorphous Toceranib was obtained by its conversion from the crystalline form during nHAp-Toc preparation”.
- The peaks at 2964 cm-1 and at 2868 cm-1 are not evident in the FTIR red line.
- From both XRD and FTIR analysis, the authors emphasize the stability of HAp in the presence of Toc, rather than identifying the interactions between these two components and if/how they bond. Please add explanations.
- L 175-176: Please explain “despite the nano-size of nHAp particles, the material has a tendency to agglomerate”. The nano-size is in fact directly associated with high specific surfaces and a great tendency to agglomerate.
- Section 3 seems more a combination of Introduction + Conclusion, rather than a Discussion part.
- Section 4.1: The pH was evaluated only after the whole phosphate-containing solution was consumed? Using a centrifuge is unlikely to separate “white fine powders”, rather than a white precipitate that is afterward dried to a powder. Also, please mention if washing was performed until neutral pH. It is quite forced to use the term “composite” for the nHAp-Toc mixtures.
- Section 4.2: Please define “appropriate amount of chemotherapeutic”. It is particularly important to know the initial, theoretical, concentration of the used drug. Moreover, how did the authors check the exact amount of Toc attached to HAp’s surface? Isn’t it possible to have just a mixture of nHAp and Toc, dispersed “in a complete cell culture medium”?
- Section 4.3: please add the analysis parameters for XRD (equipment configuration, time per step, step size, etc.).
- Section 4.4: It is not clear why the calibration curve was made based on small concentrations compared to the stock solution. Also, why did the authors choose acetic acid, instead of PBS?
- Section 4.5: Please address the ethical aspects, if necessary.
Reviewer 2 Report
It is a nice work. However, the authors should explain more clear, the benefit of nanoHAP as drug carrier, to enumerate precisely some other inorganic drug carriers.
They should write at the and short, concentrated CONCLUSIONS (as a new paragraph).
Please put also the results of the "additional UV-Vis test (see line 220).
Round 2
Reviewer 1 Report
The authors improved the manuscript in a consistent amount and can now be considered for publication.